# The Relation between Perceived and Actual Understanding and Adherence: Results from a National Survey on COVID-19 Measures in Belgium

**DOI:** 10.3390/ijerph181910200

**Published:** 2021-09-28

**Authors:** Kirsten Vanderplanken, Stephan Van den Broucke, Isabelle Aujoulat, Joris Adriaan Frank van Loenhout

**Affiliations:** 1Centre for Research on the Epidemiology of Disasters (CRED), Institute of Health and Society, Université Catholique de Louvain, 1200 Brussels, Belgium; kirsten.vanderplanken@gmail.com; 2Psychological Sciences Research Institute, Université Catholique de Louvain, 1348 Louvain-la-Neuve, Belgium; stephan.vandenbroucke@uclouvain.be; 3Centre for Health Promotion Knowledge Transfer (RESO), Institute of Health and Society, Université Catholique de Louvain, 1200 Brussels, Belgium; isabelle.aujoulat@uclouvain.be

**Keywords:** COVID-19, protective measures, adherence, perceived understanding, actual understanding, risk factors

## Abstract

To reduce the spread of COVID-19 among the population, Belgium has implemented various infection prevention and control measures over time. This study investigated the extent to which understanding of the COVID-19 measures contributed to adherence, and which personal characteristics were considered risk factors for lower adherence. It consisted of a large online survey among a sample of the population (*n* = 2008), representative of citizens of Belgium in terms of gender, age, province and socio-economic status. The survey was conducted in September 2020, and included questions on perceived and actual understanding of COVID-19 protective measures in place during that time, as well as past and future adherence to those measures. The results showed that both perceived and actual understanding contributed significantly to past as well as future adherence. Risk factors for perceived understanding included being male and belonging to a younger age group, while risk factors for actual understanding were speaking French (versus Dutch) and belonging to a lower socio-economic level. Communication about COVID-19 measures should put more focus on trying to improve the understanding of the measures, instead of only making them known, particularly for those who are less health literate and as such at risk of poor understanding.

## 1. Introduction

Since the first case of COVID-19 was detected in December 2019, human-to-human transmission of the virus has become a global concern. Until the end of July 2021, over 195 million people worldwide have been infected, of whom almost 4.2 million have died [1]. The outbreak was declared a pandemic by the World Health Organization in March 2020 [2] and countries were advised to take measures to limit transmission. Given that no vaccines were available until the end of 2020, countries had to implement non-medical infection prevention and control measures to limit the number of new infections and ease the burden on healthcare systems. These measures enforced or advised behavioural changes at the individual level, such as frequent hand washing, wearing face masks, restricting or avoiding social contact, avoiding international travel, etc. [3,4,5]. The number and type of measures implemented varied over time and between countries, and frequent adaptations were made based on the severity of the outbreak a country was experiencing at a given moment [6]. In Belgium, citizens received information and instructions for individually and community targeted protection measures through regular press conferences based on decisions by the Consultative Committee of the Federal Government. In addition, authorities at regional levels of government could implement additional, more stringent measures in response to local outbreaks.

The focus on the individual in the management of transmission risks has been a key concept in the Belgian strategy. It relies on the assumption that individuals who are properly informed are able to adhere to the applicable measures. Health literacy plays an essential role in this approach, as in order to act on information individuals are presumed to have the ability to acquire, understand, interpret and use the information provided and to subsequently implement behaviours for protecting themselves as well as others [5,7,8,9,10]. However, as again emphasised by the current pandemic, the issue of health literacy is generally underestimated in public health [8,11]. A survey conducted in Europe in 2011 found that almost one in two adults had inadequate or problematic health literacy [12]. Poor health literacy is associated with non-adherence to disease control strategies and adverse health outcomes [9,13], but may also be related to a poor motivation to adhere to protective measures against COVID-19, or even to attend to information about the pandemic. As stated by the Cognitive Mediation Model (CMM), different motivations drive people to pay more or less attention to news media and to process news information, which in the case of the pandemic could influence their willingness to perform protective behaviours [14]. Furthermore, both health literacy and health behaviour are subject to social inequality and individual differences [4,5,7]. Health literacy is considered to be an outcome of socio-demographic, individual and environmental factors, in the sense that low levels of health literacy have been associated with being male, being of older age, having a low educational level, having a low income, belonging to an ethnic minority group and living alone, amongst other characteristics [12,15]. Arguments for the fact that men have lower health literacy than women are that they have more reticence in help-seeking for health [16]. Yet although the importance of health literacy for adhering to protective measures against COVID-19 is increasingly acknowledged [9,10,11,17,18], the way in which it influences adherence is not well understood.

The current study aimed to clarify the association between the public’s level of understanding of information on protective measures and their adherence to those measures. It was hypothesised that individuals with low levels of understanding about the COVID-19 measures would adhere less to these measures and may thus contribute more to the spread of COVID-19. A distinction was made between perceived understanding and actual understanding, to also assess the extent to which respondents were able to evaluate their own level of understanding of the measures. Finally, we explored risk factors for low levels of understanding, based on demographic, socio-economic, health and information perception characteristics. The hypothesis is that characteristics identified in other health literacy studies could also be risk factors for having a reduced understanding of COVID-19 measures (e.g., being male, being of older age, having a low educational level). Insights from this study may help in developing and improving disease control policies in Belgium and beyond.

## 2. Materials and Methods

### 2.1. Data Collection

A panel-based survey was conducted among a sample representative of the Belgian adult population aged 18 to 75, aiming for a sample size of at least 2000 participants. The survey was conducted by a polling agency through their own survey tool, which collected responses from an online panel using pre-defined quotas for demographic (gender, age, province/region of residence) and socio-economic characteristics (level of education, occupational status) to ensure representativeness for the Belgian population. The survey language was Dutch or French, two official Belgian languages, which combined constitute the native language of more than 90% of the population of Belgium. Several quality controls were applied, such as the inclusion of two quality control questions in the survey (e.g., “To ensure that you complete the questionnaire correctly, please enter the number 7”) to identify and dismiss inattentive respondents, and monitoring the completion time and answer patterns, which were used to eliminate respondents who completed the survey 50% below the average time and respondents who systematically gave the same answers (e.g., always a score of ‘5’ in the Likert-scale questions).

The survey was pilot-tested during 4–7 September 2020 by the polling agency on a pre-sample of *n* = 50 who completed the survey and by the research team who asked several people to complete the survey and provide feedback on any issues and the formulation of the questions. Based on these pilot tests, minimal changes were made to the survey. The survey was fielded during 7–24 September 2020. The Belgian federal government announced new measures on 23 September. As this could impact a small number of survey questions, the responses on the questions related to the modified measures (“The social bubble is limited to five people” and “Shop with max. one other person”) were removed from files completed as of that date (*n* = 116).

### 2.2. Questionnaire

The questionnaire collected various demographic, socio-economic and health-related characteristics of the respondents. Furthermore, respondents were asked to which extent they perceived themselves to be informed about COVID-19 measures (on a scale from 0 = not at all informed to 100 = very well informed). For the protective measures that were announced on 20 August 2020 and came into force on September 1st, which were described in detail in a previously published article [19], respondents were asked to rate on Likert scales their perceived understanding (1 = “not at all”, 5 = “very well”), and their past and intended future adherence to the measures (1 = “not at all”, 5 = “completely”). They were also asked to rate on Likert scales, for different groups (politicians, experts, journalists, close contacts), to what extent these groups contributed to them being informed, and to what extent they considered the provided information to be clear and trustworthy. All scaling in the questionnaire followed the same pattern from low to high scales. Finally, respondents were asked to demonstrate their actual understanding of the protective measures by completing 10 true/false statements. The full questionnaire can be found in a previously published article [19].

### 2.3. Data Description

Educational level was re-categorised from 13 categories included in the questionnaire into 5 categories (primary or without diploma, lower secondary, upper secondary, superior short type and bachelors, long/university level superior). Overall scores were calculated for perceived understanding, past adherence and future adherence, by averaging the individual scores for each measure, resulting in a score ranging from 1 to 5. An overall score was calculated for actual understanding by summing the number of true/false statements that were answered correctly by each respondent, resulting in a score range of 0–10.

### 2.4. Data Analysis

All analyses were performed using IBM SPSS Statistics 25 (IBM Corp: Armonk, NY, USA). Descriptive overviews were presented for all variables. For continuous variables and questions with a Likert scale, average scores were calculated, as well as standard deviations. Pearson’s r correlation coefficients were calculated between three variables related to understanding of measures and being informed: perceived extent of being informed, perceived understanding and actual understanding. Risk factors for poor perceived and actual understanding were identified by undertaking multivariate linear regression. All potential risk factors were included in the initial multivariate models, and factors that were not statistically significant were removed via a backward analysis. By using multivariate versus univariate models, we retained only predictors that have a strong association with the outcome variable, and as such we reduced the chances of Type 1 errors occurring.

## 3. Results

### 3.1. Respondent Characteristics

The polling agency sent invitations for the survey to approximately 22,000 potential respondents. In total, 3257 respondents started the survey. Of these, 941 were not accepted because the quota for their socio-demographic group were reached, 177 did not complete the survey, and 131 were refused because their responses were of insufficient quality. After eliminating these, the final number of respondents was 2008. The composition of our sample was representative of the Belgian population.

The descriptive data of the respondents are presented in Table 1. About 80% (*n* = 1640) of the respondents gave a score of 60 or above to their health status on the day of completing the questionnaire (M = 72.3, SD = 18.6). Almost 90% (*n* = 1773) of respondents reported not to be dependent on someone else’s care. At the moment of completing the survey, the majority of respondents had not tested positive for COVID-19 and had not shown any symptoms (*n* = 1709).

### 3.2. Extent of Feeling Informed, Understanding and Adherence

With regard to information on prevention measures against COVID-19, approximately 90% (*n* = 1811) of the respondents gave a score of 50 or more (on a scale of 0–100) in answer to the question whether they perceived themselves to be informed (M = 74.9, SD = 21.3). In terms of perceived understanding, a high level was reported for all measures (means of 4.08 to 4.52 on a five-point scale) (Table 2). Working from home and the need to wear a mask covering the mouth and nose in busy places were best understood, while limiting the number of people admitted to an official event was least well understood. In terms of actual understanding, the average number of statements that respondents answered correctly was 5.4 out of 10 (SD = 1.8). High average scores were noted for past and intended adherence to all measures (between 4.00 and 4.68 and between 3.99 and 4.61, respectively, on a five-point scale) (Table 2). Respondents reported to have adhered the most to the measure on wearing a facemask and the least to the measure on restricting the number of personal contacts. The same trend was observed for their intended future behaviour.

### 3.3. Association between Understanding and Adherence

Positive and statistically significant associations were found between the three indicators of understanding (perceived extent of being informed, perceived understanding, actual understanding) and past and intended adherence (Table 3), confirming our hypothesis. The highest R^2^ (explained variation) was found in the model between perceived understanding and future adherence (0.249), while the lowest was found between actual understanding and future adherence (0.008).

### 3.4. Risk Factors Associated with Perceived Understanding

The associations between demographic, socio-economic, and health characteristics and information use on the one hand, and perceived understanding on the other hand, were examined by means of a multivariate model (Table 4). Men scored significantly lower than women, which confirmed the hypothesis of being male as a risk factor, and the youngest age group scored lower than older age groups, rejecting the initial hypothesis. In addition, when the information received from the different sources was considered to be clear and trustworthy, higher scores were obtained for perceived understanding, and perceiving oneself as well-informed also resulted in a higher level of perceived understanding.

### 3.5. Risk Factors Associated with Actual Understanding

Similar analyses were performed to identify potential risk factors for a poor actual understanding of COVID-19 measures. Respondents who completed the questionnaire in Dutch had a higher score than those who completed it in French. As for occupation, unemployed respondents and students scored significantly lower on actual understanding than the reference group of respondents with an occupation. In terms of educational level, people with a lower educational level scored lower on actual understanding than those with higher levels, confirming the hypothesis stated. Finally, and similar to the relationship with perceived understanding, a higher extent of perceiving oneself as well-informed resulted in a higher actual understanding.

### 3.6. Relationship between Perceived and Actual Understanding, and Perceived Extent of Being Informed

Pearson’s correlation coefficients between the different indicators measuring respondents’ understanding were relatively low, i.e., r = 0.177 between perceived and actual understanding, r = 0.362 between perceived extent of being informed and perceived understanding, and r = 0.128 between perceived extent of being informed and actual understanding, although all were statistically significant (*p* < 0.001).

## 4. Discussion

This nationwide survey involved a representative sample of 2008 citizens of Belgium and investigated their knowledge, understanding and behaviour regarding COVID-19 measures in September 2020. It confirms results from previous studies finding that understanding measures that are requested is essential to motivate people to perform appropriate protective behaviours [4]. A strong, positive association was found between all indicators measuring understanding (perceived extent of being informed, perceived understanding and actual understanding) and adherence to COVID-19 measures. This suggests that improving the general public’s understanding of the protective measures against COVID-19 is a key factor to increasing adherence and should be considered by policymakers in their communication about infection prevention and control measures.

Our study identified several characteristics that are associated with lower levels of perceived and actual understanding of the protective measures against COVID-19. Since a poor understanding of information can be considered as an element of poor health literacy, the fact that, in our study, men reported lower perceived understanding of the measures against COVID-19 than women is in line with the often-reported finding that men show lower health literacy levels [12]. Lower perceived understanding when belonging to a younger age group (18–30 years) confirms findings from other studies showing that younger people have a lower level of adherence to protective measures against COVID-19 than older people [20,21]. However, this finding seems to be specific for COVID-19, as previous research indicates higher age rather than lower age as a risk factor for low health literacy [12]. On the other hand, higher levels of perceived understanding were found among individuals who consider information from sources like politicians or health experts as clearer and more trustworthy. The latter provides an important argument for stakeholders to improve the clarity of communication about the COVID-19 measures and to develop trust between those who communicate the message and the public, since these are likely to affect adherence.

The finding that poor understanding of the protective measures is consistently and positively related to adherence also sheds light on the way in which the relationship between health literacy and adherence to measures against COVID-19 can be understood. Indeed, being able to understand information about health is one of the critical skills of health literacy, along with accessing, appraising and applying that information. Given the abundant information that is available about COVID-19, accessing information is not likely to be a problem for most people, while understanding and appraising this information may well be. As such, a limited understanding of the measures to protect against COVID-19 will not only reduce the motivation to adhere to those measures, but also to pay attention to additional or new information, and thus reduce the willingness to perform protective behaviours.

Our study also found important differences between the factors that were associated with perceived and actual understanding of the measures against COVID-19. Specifically, it was seen that French speakers had lower actual understanding than Dutch speakers, even though the measures were identical for both language groups. We cannot determine based on our study whether this finding is caused by cultural differences, communication differences or other reasons, but it would be of interest to investigate this further in future research, by, e.g., using qualitative research methods. It is also to be noted that two occupational groups had lower actual understanding of the measures than the reference group, namely the unemployed and students, and that a similar lower understanding was also seen in individuals who were less well educated. Financial deprivation and lower levels of education have been identified as risk factors for low health literacy in earlier research [12], which would explain why the unemployed and those with a lower educational level have a lower understanding of the measures against COVID-19. On the other hand, the fact that students also have less understanding could be related to their younger age. Other studies have shown that young people were found to adhere less to these measures [20,21], which could be related to a lower understanding, as shown by the results of our research. A study by the National Institute of Health in Belgium in March 2021 showed that young adults (ages 18–29) had a lower quality of life compared to older age groups, and almost 70% reported anxiety and depression symptoms [22]. If this age group seems to be most affected by the COVID-19 crisis in their mental health, it can also be a potential explanation for why they adhere less. The fact that age did not contribute significantly to our predictive model for actual understanding is likely a result of multicollinearity with occupational status.

Correlations between the perceived extent of being informed, perceived understanding and actual understanding of COVID-19 measures were relatively low, which suggests that people have difficulty assessing their own level of understanding. People may overestimate their own knowledge, and as a result be at risk of incorrect interpretation of the measures. However, while perceived and actual understanding represent two different concepts, both are strongly and significantly associated with the level of adherence to the measures. It is therefore vital to improve the knowledge about and understanding of COVID-19 measures among the groups that are identified as being particularly at risk for having a lesser knowledge and understanding. Nonetheless, the explained variation (R^2^) in each of the models was lower than 0.25, indicating that other factors exist, which are not described in this study, which are associated with adherence.

This study had some limitations. While the sample represents the Belgian population well in terms of gender, age (adult population aged up to 75), region and socio-economic status, those belonging to the lowest socio-economic group were slightly under-represented. This is partly due to the use of our survey method, as people with a lower socio-economic status are more often deprived of access to the internet, may have more difficulty understanding a survey, or are less likely to participate in surveys due to a feeling of inferiority (“my opinion does not matter”). A second limitation is that certain groups are by definition underrepresented, such as young people under 18, people from the informal sector (e.g., asylum seekers, sex workers), and people who do not speak the survey language (e.g., migrants, expats). Finally, due to our large sample size, there is a possibility of Type 1 errors (false positive relationships) occurring in the analyses on the risk factors for low understanding. However, we reduced the chances of this happening by having performed multivariate analyses, versus univariate analyses, which retains only predictors in the model with strong associations with the outcome variable. Despite these limitations, however, we believe that our findings contribute to clarifying the association between the public’s level of understanding of information on protective measures against COVID-19 and their adherence to those measures.

## 5. Conclusions

The findings from this study confirm the hypothesis that a poor understanding of measures to protect against COVID-19 leads to low adherence. Moreover, several factors were identified that impact on this understanding, including male gender, young age, lower education, being French-speaking rather than Dutch-speaking, and belonging to a lower socio-economic segment of the population. Communication about COVID-19 measures can address these issues by putting more focus on trying to improve the understanding of the measures, instead of only making them known, particularly for those who are less health literate and as such at risk of poor understanding. Communication efforts should thus help people understand the reasons for the measures and how they can be put into practice. This should also include actively using the communication channels that are used by the identified groups at risk of poorer understanding. Nonetheless, understanding represents only one aspect that influences adherence, and a deeper investigation in other factors is required in order to improve adherence levels in the overall population.

## Figures and Tables

**Table 1 ijerph-18-10200-t001:** Demographic, socio-economic and health characteristics.

Characteristic	N	%	Characteristic	N	%
**Gender**			**Educational level**		
Male	983	49.0	Primary or without diploma	62	3.1
Female	1024	51.0	Lower secondary	240	12.0
Other	1	0.0	Upper secondary	810	40.3
**Age**			Superior short type and bachelors	420	20.9
18–30 years	407	20.3	Long/university level superior	471	23.5
31–45 years	472	23.5	**Occupation**		
46–60 years	522	26.0	Yes	920	45.8
61–75 years	557	27.7	No, incapacitated to work	161	8.0
76 years and over	50	2.5	No, pre-pension	33	1.6
**Province**			No, pension	530	26.4
Antwerp	320	15.9	No, unemployed	80	4.0
Flemish Brabant	184	9.2	No, student	180	9.0
Limburg	154	7.7	No, homemaker	88	4.4
West Flanders	219	10.9	No, never or not yet worked	16	0.8
East Flanders	263	13.1	**Net annual household income**		
Brussels Capital Region	206	10.3	Less than EUR 15,000	164	8.2
Walloon Brabant	74	3.7	Between EUR 15,000 and 29,999	612	30.5
Hainaut	250	12.5	Between EUR 30,000 and 44,999	534	26.6
Liège	206	10.3	More than 45,000	319	15.9
Luxembourg	50	2.5	I do not know	379	18.9
Namur	82	4.1	**Native language**		
**Region**			Dutch	1072	53.4
Flanders	1140	56.8	French	793	39.5
Wallonia	662	33.0	English	29	1.4
Brussels	206	10.3	Other EU language	65	3.2
**Household composition**			Arabic	19	0.9
Alone without children	474	23.6	Russian	8	0.4
Alone with children	135	6.7	Turkish	6	0.3
Couple without children	655	32.6	Sub-Saharan African language	4	0.2
Couple with children	494	24.6	**Contracted COVID-19**		
With parents	229	11.4	Not tested positive and no COVID-19 symptoms	1709	85.1
Live together/share a flat	21	1.0	Not tested positive but had COVID-19 symptoms	199	9.9
**Dependency on care**			Tested positive but without COVID-19 symptoms	42	2.1
Never	1773	88.3	Tested positive for COVID-19 symptoms and hospitalised	1	0.0
Less than once a month	68	3.4			
1–3 times a month	81	4.0	Tested positive for COVID-19 symptoms but no hospitalisation	26	1.3
1–3 times a week	41	2.0			
More than 3 times a week	45	2.2	Do not know if tested positive for COVID-19	33	1.6

**Table 2 ijerph-18-10200-t002:** Understanding of and adherence to COVID-19 measures.

	Understand	Adhered Past	Adhere Future
Measure	Mean (SD)	Mean (SD)	Mean (SD)
Social bubble limited to 5	4.11 (1.13)	4.00 (1.26)	3.99 (1.29)
Private events limited to 10	4.10 (1.14)	4.42 (1.04)	4.27 (1.11)
Official events limited to 200 (indoors) or 400 (outdoors)	4.08 (1.17)	4.58 (0.90)	4.52 (0.93)
Homeworking strongly recommended	4.52 (0.86)	4.08 (1.30)	4.16 (1.25)
Shop with max. one other person	4.38 (1.01)	4.55 (0.93)	4.45 (1.01)
Wearing a facemask in public spaces	4.50 (0.91)	4.68 (0.74)	4.61 (0.83)
Travel form	4.25 (1.07)	4.40 (1.05)	4.45 (1.00)
Travel zones	4.11 (1.14)	4.39 (1.03)	4.48 (0.99)

**Table 3 ijerph-18-10200-t003:** Association between understanding and adherence, using univariate models.

Understanding	Implementation					
	Past			Future		
	B-Value (CI)	*p*-Value	R^2^	B-Value (CI)	*p*-Value	R^2^
Informedness (/100)	0.67 (0.52; 0.82)	<0.001	0.038	0.87 (0.71; 1.03)	<0.001	0.053
Perceived understanding	0.44 (0.40; 0.47)	<0.001	0.233	0.50 (0.46; 0.54)	<0.001	0.249
Actual understanding	0.04 (0.02; 0.06)	<0.001	0.010	0.04 (0.02; 0.06)	<0.001	0.008

**Table 4 ijerph-18-10200-t004:** Demographic, socio-economic and health characteristics associated with understanding.

Characteristic		Perceived Understanding		Actual Understanding	
		B-Value (CI)	*p*-Value	B-Value (CI)	*p*-Value
Intercept		2.5 (2.4; 2.7)	<0.001	4.9 (4.6; 5.3)	<0.001
Language					
	French			−0.5 (−0.7; −0.4)	<0.001
	Dutch			Ref	Ref
Gender					
	Male	−0.1 (−0.2; −0.1)	<0.001		
	Female	Ref	Ref		
Age			<0.001		
	18–30 years	Ref	Ref		
	31–45 years	0.2 (0.1; 0.3)	<0.001		
	46–60 years	0.2 (0.1; 0.3)	<0.001		
	61–75 years	0.3 (0.2; 0.4)	<0.001		
	76 years and over	0.1 (−0.1; 0.3)	0.212		
Occupation					0.003
	No, incapacitated to work			0.0 (−0.3; 0.3)	0.879
	No, prepension			0.0 (−0.6; 0.7)	0.900
	No, pension			0.1 (−0.1; 0.3)	0.453
	No, unemployed			−0.4 (−0.9; 0.0)	0.034
	No, student			−0.4 (−0.7; −0.1)	0.003
	No, homemaker			0.3 (−0.1; 0.7)	0.190
	No, never or not yet worked			0.9 (−0.1; 1.8)	0.065
	Yes			Ref	Ref
Educational level					<0.001
	Primary or without diploma			−0.9 (−1.3; −0.4)	<0.001
	Lower secondary			−0.4 (−0.7; −0.2)	0.002
	Upper secondary			0.1 (−0.1; 0.3)	0.562
	Superior short type and bachelors			0.2 (0.0; 0.4)	0.120
	Long/university level superior			Ref	Ref
Clear information		0.1 (0.1; 0.2)	<0.001		
Trustworthy		0.2 (0.1; 0.2)	<0.001		
Extent of perceiving themselves informed (/100)		0.9 (0.8; 1.1)	<0.001	1.0 (0.6; 1.4)	<0.001

Perceived understanding model: R^2^ = 0.233; adjusted R^2^ = 0.230. Actual understanding model: R^2^ = 0.061; adjusted R^2^ = 0.055.

## Data Availability

The data used for this study can be accessed through the UCLouvain Dataverse, using the following https://doi.org/10.14428/DVN/VJUPXI.

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
