# Peer review of "The Relation between Perceived and Actual Understanding and Adherence: Results from a National Survey on COVID-19 Measures in Belgium"

_ijerph, 2021, doi:10.3390/ijerph181910200_

Round 1
Reviewer 1 Report
Well written introduction with a strong rationale for the study.
Could further clarification be made about the measure that came into place in September 2020 (lines 99- 102). These lines state the responses to these questions were removed. However, in lines 107 – 110, the authors talk about responses to these measures. It’s a little confusing as if respondents knew about the measures, in advance, then the date of implementation should not matter? This needs further clarification.
Clarification confirming that all scaling followed the low to high (increasing from zero) scales as described in section 2.2 would be helpful.
Lines 238 to 243. Did the study by the National Institute of Health explicitly link anxiety and depression symptoms to Covid-19 as suggested in lines 241 – 242?
Helpful article that provides evidence to support inclusive and varied public health announcements /information sharing, which supports current dialogue regarding informing more marginalised communities about vaccinations.
Author Response
Well written introduction with a strong rationale for the study.
Could further clarification be made about the measure that came into place in September 2020 (lines 99- 102). These lines state the responses to these questions were removed. However, in lines 107 – 110, the authors talk about responses to these measures. It’s a little confusing as if respondents knew about the measures, in advance, then the date of implementation should not matter? This needs further clarification.
We understand the reviewer’s confusion. The measures that we included in the survey were announced on the 20th of August and applied for the month of September. However, on the 23rd of September, a modification took place in some of the measures, including two which were included in our survey (‘The social bubble is limited to 5 people’ and ‘Shop with max. one other person’). We have adjusted the text in the manuscript to clarify this (Lines 112-113 and 119-120).
Clarification confirming that all scaling followed the low to high (increasing from zero) scales as described in section 2.2 would be helpful.
We have incorporated the reviewer’s suggestion (Lines 126-127). Furthermore, we noticed a small error in the text, namely that our Likert scales options were between 1 and 5, instead of between 0 and 5 as originally. We have corrected this in the manuscript (Lines 122-123).
Lines 238 to 243. Did the study by the National Institute of Health explicitly link anxiety and depression symptoms to Covid-19 as suggested in lines 241 – 242?
The symptoms were indeed not linked to being infected with COVID-19. We understand the confusion that might arise as a result of our phrasing, and because of that we have replaced ‘COVID-19’ in the next sentence by ‘the COVID-19 crisis’, which is what we intended to say initially (Line 275).
Helpful article that provides evidence to support inclusive and varied public health announcements /information sharing, which supports current dialogue regarding informing more marginalised communities about vaccinations.
Reviewer 2 Report
I believe this study has strong potential given the high relevance of the topic and the well-implemented survey, but major changes are necessary.
In the “Introduction” section, the authors need to back up their study with theoretical underpinnings. Specifically, they need to bring in a theory which explains people’s motivations on health literacy and why certain demographic groups differ in these motivations. Based on this theory, they also need to develop hypotheses for the study. By doing so, the article can add to the academic literature on health literacy and give the research a broader relevance.
In the “data collection” section, the exact number of quality control questions needs to be specified. The authors should also explain more specifically what is meant by “respondents who systematically gave the same answers” (line 94 on page 2). In addition, the exact measures implemented by the Belgian government at the beginning of September, 2020, need to be described. Finally, the authors should confirm that these measures were put in place on 9/1/21.
The “results” section is very confusing and needs to be structured in a way which the reader can more easily follow. I would suggest that the authors divide it into 5 sections: (1) perceived themselves to be informed and adherence, (2) perceived understanding and adherence, (3) understanding and adherence, (4) risk factors associated with perceived understanding, and (5) risk factors associated with understanding. The current section on “relationship between perceived and actual understanding, and perceived extent of being informed” is not as central to the study and should either be deleted or placed at the end.
Another important point about the results section has to do with the large sample size. Given this sample size, the probability statistics are less relevant given the chances of a Type 1 error. Thus the bulk of the discussion should be on the beta and correlation figures. If these are small, the importance of a given result is diminished. The authors do make this point at different times, but it needs to be applied more consistently throughout the entire section.
The results section should also mention whether hypotheses (to be added per my other comment) are supported or rejected.
Once all the above changes are made, the discussion section should be organized to reflect the new structure of the results section. Also, be careful not to infer causation, rather than association, in the discussion.
Author Response
I believe this study has strong potential given the high relevance of the topic and the well-implemented survey, but major changes are necessary.
In the “Introduction” section, the authors need to back up their study with theoretical underpinnings. Specifically, they need to bring in a theory which explains people’s motivations on health literacy and why certain demographic groups differ in these motivations. Based on this theory, they also need to develop hypotheses for the study. By doing so, the article can add to the academic literature on health literacy and give the research a broader relevance.
We thank the reviewer for this suggestion. We have added literature on the Cognitive Mediation Model, which describes motivations on why people pay less or more attention to news (Lines 61-66). In addition, we have added a study that provides an argumentation for why men are considered to have lower health literacy than women (Lines 72-74) and we have added an additional hypothesis in relation to this (Lines 85-88).
In the “data collection” section, the exact number of quality control questions needs to be specified. The authors should also explain more specifically what is meant by “respondents who systematically gave the same answers” (line 94 on page 2). In addition, the exact measures implemented by the Belgian government at the beginning of September, 2020, need to be described. Finally, the authors should confirm that these measures were put in place on 9/1/21.
The questionnaire contained two quality control questions, which we have added in the text (Line 100). We also added more information on respondents who systematically gave the same answers (line 105). A complete overview of measures that applied in September 2020 was included in a previously published article, which we reference in the text in lines 120-121. We also clarify that these measures were put in place on the 1st of September 2020 (Line 120).
The “results” section is very confusing and needs to be structured in a way which the reader can more easily follow. I would suggest that the authors divide it into 5 sections: (1) perceived themselves to be informed and adherence, (2) perceived understanding and adherence, (3) understanding and adherence, (4) risk factors associated with perceived understanding, and (5) risk factors associated with understanding. The current section on “relationship between perceived and actual understanding, and perceived extent of being informed” is not as central to the study and should either be deleted or placed at the end.
Following the reviewer’s suggestion, we have made several modifications to the results section:
- We have added two headings, to clarify the flow of information (Lines 183 and 210).
- In various places, we have indicated more clearly to which of the three indicators we were referring (extent of feeling informed, perceived understanding, or actual understanding).
- We have moved the section “relationship between perceived and actual understanding, and perceived extent of being informed” to the end of the Results.
The order is now following the order proposed by the reviewer. We have just decided not to split up proposed sections 1, 2 and 3 into separate headings, as we feel this split is a bit artificial, and it will make it challenging to find an appropriate place for the description of adherence (as this cannot be repeated under each heading).
Another important point about the results section has to do with the large sample size. Given this sample size, the probability statistics are less relevant given the chances of a Type 1 error. Thus the bulk of the discussion should be on the beta and correlation figures. If these are small, the importance of a given result is diminished. The authors do make this point at different times, but it needs to be applied more consistently throughout the entire section.
This issue relates mostly to the results in Table 4, where there is some likelihood of a Type 1 error due to a large sample size. However, we feel the chances of this occurring are reduced due to fact that we undertook multivariate regression, which eliminates factors with a weak association. We have added text in the Methods (Lines 149-151) and Discussion (Lines 299-303), to underline this point.
The results section should also mention whether hypotheses (to be added per my other comment) are supported or rejected.
We have indeed added information on whether hypotheses were confirmed or rejected at several places in the Results section (Lines 186-187, 202-203, 217-218), as well as in the Discussion (Lines 248-249).
Once all the above changes are made, the discussion section should be organized to reflect the new structure of the results section. Also, be careful not to infer causation, rather than association, in the discussion.
We consider it important that the order of the topics in the Discussion section not only follows the same order as the results, but that the topics are also ranked in terms of prioritisation (starting with the main findings of the paper). We feel that the current order is a good representation of both. We have also made sure not to infer causation in the text, or only if causation is ‘suggested’, which we feel is the case now in the whole manuscript.
Reviewer 3 Report
Please, see the attachment

Author Response
Thank you for giving me the opportunity to read and comment a report “The Relation between Perceived and Actual Understanding and Adherence: Results from A National Survey on COVID-19 Measures in Belgium”, by Vanderplanken K,et al.
In the reviewed manuscript, the association between the public’s level of understanding of information on protective measures for COVID-19 and their adherence to those measures have been evaluated.
This paper is well written, correctly structured with a suitable research concept, the study limitations are addressed, and it is of relevance to readers of the journal.
Only the modification shown below is suggested:
- It would be appropriate that the categorical variables represented in terms of frequency distribution are accompanied by their corresponding confidence intervals.
We thank the reviewer for these encouraging words. With respect to the suggestion: we are not exactly sure what the comment is referring to. The only place in the manuscript in which categorical variables are displayed is Table 1. However, since the proportions displayed there represent exact fractions of the sample (e.g. the proportion of men within the sample was exactly 49.0%), there are no confidence intervals associated to them. In case the reviewer is referring to something else, his or her clarification on this would be very helpful.
Round 2
Reviewer 2 Report
Thank you for your work on improving the article. I believe you addressed my major issues. The one change still needed is to discuss (in the Discussion section) the implications of your results for the Cognitive Mediation Model. In this way, you can show how your article has some bearing on the theoretical literature in your area.
Author Response
We thank the reviewer for this suggestion. We have added an additional paragraph in the Discussion (Lines 252-262, marked in yellow), in which we address this